# A Novel Metric for Alzheimer’s Disease Detection Based on Brain Complexity Analysis via Multiscale Fuzzy Entropy

**DOI:** 10.3390/bioengineering11040324

**Published:** 2024-03-27

**Authors:** Andrea Cataldo, Sabatina Criscuolo, Egidio De Benedetto, Antonio Masciullo, Marisa Pesola, Raissa Schiavoni

**Affiliations:** 1Department of Engineering for Innovation, University of Salento, 73100 Lecce, Italy; antonio.masciullo@unisalento.it (A.M.); raissa.schiavoni@unisalento.it (R.S.); 2Department of Electrical Engineering and Information Technology (DIETI), University of Naples Federico II, 80125 Naples, Italy; sabatina.criscuolo@unina.it (S.C.); egidio.debenedetto@unina.it (E.D.B.); marisa.pesola@unina.it (M.P.)

**Keywords:** Alzheimer’s disease, multiscale fuzzy entropy, brain complexity, entropy, AD diagnosis, EEG, measurements

## Abstract

Alzheimer’s disease (AD) is a neurodegenerative brain disorder that affects cognitive functioning and memory. Current diagnostic tools, including neuroimaging techniques and cognitive questionnaires, present limitations such as invasiveness, high costs, and subjectivity. In recent years, interest has grown in using electroencephalography (EEG) for AD detection due to its non-invasiveness, low cost, and high temporal resolution. In this regard, this work introduces a novel metric for AD detection by using multiscale fuzzy entropy (MFE) to assess brain complexity, offering clinicians an objective, cost-effective diagnostic tool to aid early intervention and patient care. To this purpose, brain entropy patterns in different frequency bands for 35 healthy subjects (HS) and 35 AD patients were investigated. Then, based on the resulting MFE values, a specific detection algorithm, able to assess brain complexity abnormalities that are typical of AD, was developed and further validated on 24 EEG test recordings. This MFE-based method achieved an accuracy of 83% in differentiating between HS and AD, with a diagnostic odds ratio of 25, and a Matthews correlation coefficient of 0.67, indicating its viability for AD diagnosis. Furthermore, the algorithm showed potential for identifying anomalies in brain complexity when tested on a subject with mild cognitive impairment (MCI), warranting further investigation in future research.

## 1. Introduction

Alzheimer’s disease (AD) is a progressive neurodegenerative brain disorder that mainly leads to a decline in memory and cognitive functioning, affecting daily activities. AD especially affects individuals over the age of 65 and is recognized by the World Health Organization as the most common form of dementia, representing a global public health priority [1]. The distinguishing characteristics of AD include the accumulation of abnormal protein deposits, such as β-amyloid plaques and Tau tangles, in the brain [2]. These pathological changes disrupt communication between nerve cells and result in a gradual deterioration of brain tissue [2]. Genome-wide analyses have identified numerous genetic variants associated with AD. Notably, the apolipoprotein E (ApoE) gene, particularly its ϵ4 allele, is the most substantial and consistently observed genetic risk factor for the development of AD among these variants.

Nowadays, some drugs can slow the progression of AD, but a definitive cure does not yet exist [3]. Therefore, methods capable of diagnosing, detecting, and monitoring AD at each stage are needed [4]. For this purpose, different tools have been employed, such as neuroimaging techniques (i.e., positron emission tomography—PET, magnetic resonance imaging—MRI), blood biomarkers, and cognitive questionnaires (i.e., the mini-mental state examination—MMSE) [2]. However, these methods exhibit some limitations. PET and MRI present low temporal resolution, high costs, and invasiveness [5]. Whereas, MMSE could be affected by biases, intentional behavior, or the experience of neurologists [6].

In recent years, there has been a growing interest in exploring electroencephalography (EEG) signals for AD detection. EEG is a non-invasive and low-cost technique for evaluating brain electrical activity by placing electrodes on the scalp [7,8]. It has a high temporal resolution, allowing analysis of the transient and temporal dynamic nature of brain oscillations in both normal and abnormal conditions [9]. As a consequence, EEG-based biomarkers could offer a promising solution for AD diagnosis and monitoring.

Indeed, several studies have tried to develop EEG-based biomarkers for AD [4] showing alterations in EEG dynamics. Specifically, a reduction in high-frequency activities (α, 8–13 Hz, and β, 13–30 Hz) and an increase in low-frequency activities (δ, 0.5–4 Hz, and θ, 4–8 Hz) have been observed in AD patients compared to healthy subjects [4,10]. Consequently, the ratio between δ and α power of EEG signals and the ratio between θ and α power have been applied for AD diagnosis [4].

Nevertheless, traditional approaches based on frequency and temporal analysis of the EEG assume the neural system is stationary, neglecting the non-stationary neuronal processes [11,12]. Instead, brain processes are neither purely regular nor totally random [13]. Therefore, complexity measures for EEG signals may offer a novel perspective on understanding physiological processes in AD [4,14].

Although several studies have used different measures of EEG complexity over the years to assess the status of AD patients [4], a lack of standardized approaches and normative values persist in the assessment of the complexity of EEG signals.

Based on these considerations, the aim of this work is to develop an objective metric for AD detection, exploiting the complexity of EEG signals through Multiscale Fuzzy Entropy (MFE) [15]. The ultimate goal is to provide standardized references for AD diagnosis to support clinicians.

In more detail, the initial step was to define the typical brain entropy patterns in the various frequency bands of the EEG signal for healthy subjects (HS) and patients with AD. As expected from the results of the previous study [9], HS and AD patients constitute two different distributions. Using statistics on these distributions, specifically the mean and uncertainty of the mean, a novel metric was developed for AD detection. This metric can estimate the probability of a subject under test (SUT) being healthy or affected by AD. The metric was developed by considering 40 HS and 40 patients with AD, and then the algorithm was tested on 12 HS and 12 AD subjects, showing promising results in terms of test diagnosis performance. Specifically, the novel metric achieved an accuracy of 83%, with a diagnostic odds ratio of 25 and a Matthews correlation coefficient of 0.67. More details related to the implementation procedure will be provided in the following sections. Furthermore, the developed algorithm was tested on an individual diagnosed with mild cognitive impairment (MCI), a transitional stage between the typical age-related decline in cognitive abilities and the more severe impairment seen in dementia [16]. MCI is associated with an elevated risk of dementia, particularly AD [16]. In this instance, the proposed method shows its potential for detecting anomalies that necessitate warrant further investigation in future studies.

In light of the obtained findings, this novel metric has the potential to detect AD with high accuracy and reliability, using only a few minutes of brain EEG recording. It is an objective, easy-to-use, and low-cost tool that can support clinicians in the diagnosis phase, help prevent the progression of AD, and improve the quality of life of patients and their caregivers.

This paper is structured as follows. Section 2 discusses related works on the analysis of complexity in subjects with AD, specifically by using multiscale fuzzy entropy. Section 3 outlines the proposed method, while Section 4 explains in detail the algorithm implementation with the employed dataset. Finally, Section 5 reports the results obtained by applying the proposed algorithm on test subjects. Conclusions and future work are outlined in Section 6.

## 2. Background

Signal complexity quantifies the degree of irregularity and variability associated with the informational content embedded within the temporal or spatial dimension of a signal [17,18]. Specifically, two major attributes of a complex system are *predictability* and *regularity*, which provide complementary insights into its dynamic behavior [19]. In particular, predictability focuses on the temporal–spatial evolution of a system’s state and is measured with fractal metrics [20], while regularity analyzes the presence of repeated patterns in a time series. In the context of brain complexity, measures of regularity, such as entropy metrics, are frequently employed to assess the presence of repeated patterns in the time series [21].

Several studies in the literature provide compelling evidence of substantial alterations in entropy values within individuals with AD compared to control counterparts [4,19]. This suggests that the complexity of EEG signals provides detailed insights into neural mechanisms that other measurement approaches may not capture, resulting in a high potential for diagnostics and early detection of AD. In the last few years, multiscale formulations of entropy showed good sensitivity to AD severity, reflecting a decline in complexity from moderate to severe stages and including cases of mild cognitive impairment (MCI) [21,22]. Various studies have utilized traditional formulations such as Shannon entropy, permutation entropy, and approximate entropy, with sample entropy currently standing as a widely adopted approach, especially in its multiscale formulation (MSE) [23]. In a study by Mizuno et al. [24], a decline in MSE was observed at small time scales in frontal regions, accompanied by an increase in brain complexity at larger time scales across various brain areas. In a separate study by Sun et al. [14], statistically significant differences in MSE were revealed among the temporal, occipito–parietal, and right frontal lobes when comparing individuals with AD, MCI, and healthy controls. Specifically, healthy individuals exhibited higher entropy than both MCI and AD patients at short-scale factors, while the reverse trend was observed at long-scale factors. However, recent focus has shifted toward fuzzy entropy and multiscale fuzzy entropy (MFE) [25], considered as a natural evolution of the sample entropy methodology [26]. In EEG analysis, fuzzy entropy assesses signal similarity across the entire time series through a membership function, assigning a continuous value between 0 and 1 to denote the degree of pattern matching. In detail, at a single time scale, fuzzy entropy quantifies the conditional probability that patterns identified for *m* points persist in similarity for the subsequent (m+1) points. It is expressed as follows:(1)$$FuzzyEn\left(m,n,r\right)=\ln\Phi^{m}\left(n,r\right)-\ln\Phi^{m+1}\left(n,r\right),$$
where $\Phi^{m}$ is a function determined by the mean of all unique sequences of length *m*, while *n* and *r* denote two parameters used to adjust the smoothness of a chosen exponential membership function, fulfilling two criteria: continuity and maximization of self-similarity [9,12]. This computation can be iterated across multiple time scales by transforming the original time series x(n), according to the following formula:(2)$$y_{s}\left(t\right)=\sum_{i=j}^{j+s-1}x\left(i\right),\;\mathrm{for}\;1\leq j\leq N-s+1,$$
where $y_{s}\left(t\right)$ represents the new time series for the MFE calculation at the *s*-th scale factor. In a study by Su et al. [22], MFE on 15 scale factors was employed for the early diagnosis of patients with MCI. The results suggested heightened sensitivity in the prefrontal lobe, with normal controls exhibiting greater MFE values in the Fp1 and Fp2 channels compared to individuals with cognitive decline.

Recently, in [9], the authors demonstrated that MFE has a strong ability across diverse time scales to distinguish between individuals with AD and healthy controls, as detailed in Table 1.

Notably, a trend inversion in entropy computation within the 0.5Hz to 30Hz frequency range was observed. On the other hand, when assessing individual frequency bands, AD subjects showed higher complexity in slow frequency bands (δ and θ) compared to healthy controls, while lower complexity values were observed in fast frequency bands (α and β). Statistical analysis revealed a noteworthy level of significance in the separability between the two groups across nearly all channels and scale factors. Ultimately, MFE values for both groups exhibited robust clustering, as evidenced by silhouette scores exceeding 0.60 for all frequency bands.

Building upon promising findings and considerations toward methodological standardization, preprocessing operations, such as resampling the whole dataset to the lowest sampling frequency of 250 Hz and applying channel-wise normalization for all EEG recordings [9], were adopted in the following work. In particular, the sampling frequency is closely tied to the temporal scale of pattern detection and, consequently, the complexity of the signal under examination. On the other hand, the amplitude of EEG traces is primarily influenced by factors unrelated to the presence or absence of the disease. Hence, procedural steps such as resampling and normalization are crucial for facilitating the comparison of entropy values derived from different subjects, acquired even with varying instrumentation, and spanning multiple years.

Therefore, the entropy-based methodology introduced in the present study, elaborated on in Section 3, is grounded in prior observations and considerations on the paramount significance of minimizing variability and unifying the procedure.

## 3. Proposed Method

As mentioned in Section 2 and based on the results reported in [9], it is possible to state that the brain complexity of subjects with AD shows characteristics that are easily distinguishable from that of HS subjects. These distinctions are found both in the different EEG bands and channels but also as the scaling factors vary according to the adopted multiscale approach.

Starting from these considerations, this study proposes a metric for AD detection, exploiting the differences in the MFE values between HS and AD subjects. To enhance the understanding of the proposed method, a conceptual diagram that outlines the main steps has been reported in Figure 1. As shown in (step i), assuming that AD and HS are distinct distributions, the proposed method involves the definition of two reference intervals based on entropy trends. This approach leverages both the mean and associated uncertainties with a 95% confidence interval from the respective distributions. The method proceeds with an iteration procedure based on scaling factors, channels, and bands (step ii) for the generation of standardized scales (step iii). These scales include intervals related to AD and HS conditions that vary in size and proximity, which in turn will influence the final output according to a certain weight (step iv). In this regard, the intervals in which there is less uncertainty and greater separability between distributions were chosen to be valued appropriately, penalizing those with reduced separability. Subsequently, the reference intervals are used to compare the entropy value of a subject under test (SUT) (step v). This comparison, conducted for each EEG scaling factor, channel, and band, yields values (ϕ values) that are employed to compute two percentage indices, namely $I_{AD}$ and $I_{HS}$, indicating the proximity of the SUT to either the AD or healthy condition (step vi).

The conceptual diagram will be expanded upon in Section 4, where each aspect of the implementation is explored in detail.

## 4. Implementation

In this section, the implementation of the proposed algorithm is outlined, beginning with an overview of the datasets employed for establishing reference intervals and evaluating the proposed metric. Figure 2 shows a flowchart of the proposed methodology.

### 4.1. Dataset

In the following, initially, the datasets from which the EEG traces for HS and subjects with AD were selected are outlined. Subsequently, the process of obtaining the dataset for the development and testing of the proposed algorithm is explained.

CAUEEG dataset: The Chung-Ang University Hospital EEG (CAUEEG) dataset [27] is a collection of EEG signals recorded from 2012 to 2020 including subjects with different AD stages and healthy controls. Each EEG trace contains annotations regarding the subject’s age, event descriptions (closed and opened eyes, visual stimulation), and the diagnosis decided by the medical personnel. The data were acquired with an EEG system at a sampling frequency of 200 Hz, following the International 10–20 system (electrode locations: Fp1, F3, C3, P3, O1, Fp2, F4, C4, P4, O2, F7, T3, T5, F8, T4, T6, FZ, CZ, and PZ) with linked earlobe referencing. For the present study, 35 HS and 35 subjects with AD were selected.TUH EEG dataset: The Temple University Hospital (TUH) EEG Corpus is a publicly available collection of EEG signals acquired from 2002 to 2017 [28]. Each EEG trace is accompanied by a report with the diagnosis and the medical history of the subject, including personal details (age and gender). The EEG traces of this dataset were acquired with different EEG systems at different sampling frequencies (250 Hz, 256 Hz, 400 Hz, and 512 Hz), whereas the electrode locations followed the same 19 electrodes as the CAUEEG dataset. In the context of this study, 17 EEG signals of HS subjects and 17 EEG records of AD patients were selected.

As described above, 70 subjects (35 HS and 35 AD) from the CAUEEG dataset and 34 subjects (17 HS and 17 AD) from the TUH dataset were considered, for a total of 104 individuals (52 AD patients and 52 HS). The subjects present average age of (77.9 ± 6.7) years. Since the data have different sampling frequencies, the EEG signals were resampled at 200 Hz (the minimum found in the used data). This resampling procedure was employed to establish standardization and uniformity among signals, maintaining an equivalent number of samples for each signal. This pre-processing phase is necessary to ensure a consistent interpretation of MFE, enabling the analysis of comparable patterns for the same scale factor across all subjects.

The EEG data were pre-processed as follows:*epoching*: for each subject, 15 epochs of 3-second clean EEG signals in the eyes-closed condition were selected;*normalization*: to mitigate inter-subject variation and enhance the independence of the outcomes, the normalization of EEG signals was conducted. Following [9], a channel-by-channel scaling technique was employed for each EEG signal:
(3)$$z_{norm,i}=\frac{z_{i}-z_{min,i}}{z_{max,i}-z_{min,i}}\;\cdot \;\left(H_{max}-H_{min}\right)+H_{min}$$
where *i* refers to channels (from 1 to 19), $H_{max}$ and $H_{min}$ are the amplitude range (from −10 to 10), $z_{max,i}$ and $z_{min,i}$ represent the median values of all maxima and minima values computed for each epoch.*filtering*: EEG data were filtered with a band-pass filter between 0.5 Hz to 30 Hz. Then, δ (0.5 Hz to 4 Hz), θ (4 Hz to 8 Hz), α (8 Hz to 13 Hz), and β (13 Hz to 30 Hz) bands were considered. A finite impulse response (FIR) filter, with an order corresponding to the number of samples in a single epoch, was employed. Consequently, the initial epoch related to the transient was excluded from each time series, resulting in a total of 14 epochs considered for each subject.

The subjects were carefully selected to have a well-balanced group of HS and AD patients in terms of age. The AD patients were diagnosed with a high-severity degree of AD, without other pathologies. Moreover, despite the utilization of different datasets and different EEG signal acquisition equipment, the MFE data from the different datasets are comparable due to the careful pre-processing step. Indeed, a meticulous manual inspection and removal of noisy epochs on closed-eye signals were performed to refine the data quality further. This preprocessing phase was a cornerstone of this study since this approach standardized the EEG signals before the extraction of MFE, ensuring that any differences in entropy values reflected variations in neural complexity rather than in acquisition methodologies. As a matter of fact, the MFE values calculated from different datasets were directly comparable. Moreover, the consistency of MFE values obtained from different datasets reinforces the validity of MFE as a reliable biomarker, irrespective of experimental conditions. On such basis, after this pre-processing, the data were partitioned to generate reference distributions and test sets for the validation of the metric. Specifically, the reference set presented 40 AD and 40 HS subjects (13 AD and 13 HS subjects from the TUH dataset and 27 AD and 27 HS subjects from the CAUEEG dataset). On the other hand, the residual data (12 HS and 12 AD selected by choosing the remaining 4 AD and 4 HS subjects from the TUH dataset and 8 AD and 8 HS subjects from the CAUEEG dataset) were employed for the validation of the proposed metric.

### 4.2. Algorithm Implementation

As described in Section 3, the basic concept of the developed algorithm is to report each entropy value of the SUT on a specific scale in order to obtain, through the use of appropriate weights and processing, a final diagnostic metric. The different steps of the algorithm implementation are summarized in the flowchart in Figure 2.

**Figure 2 bioengineering-11-00324-f002:**
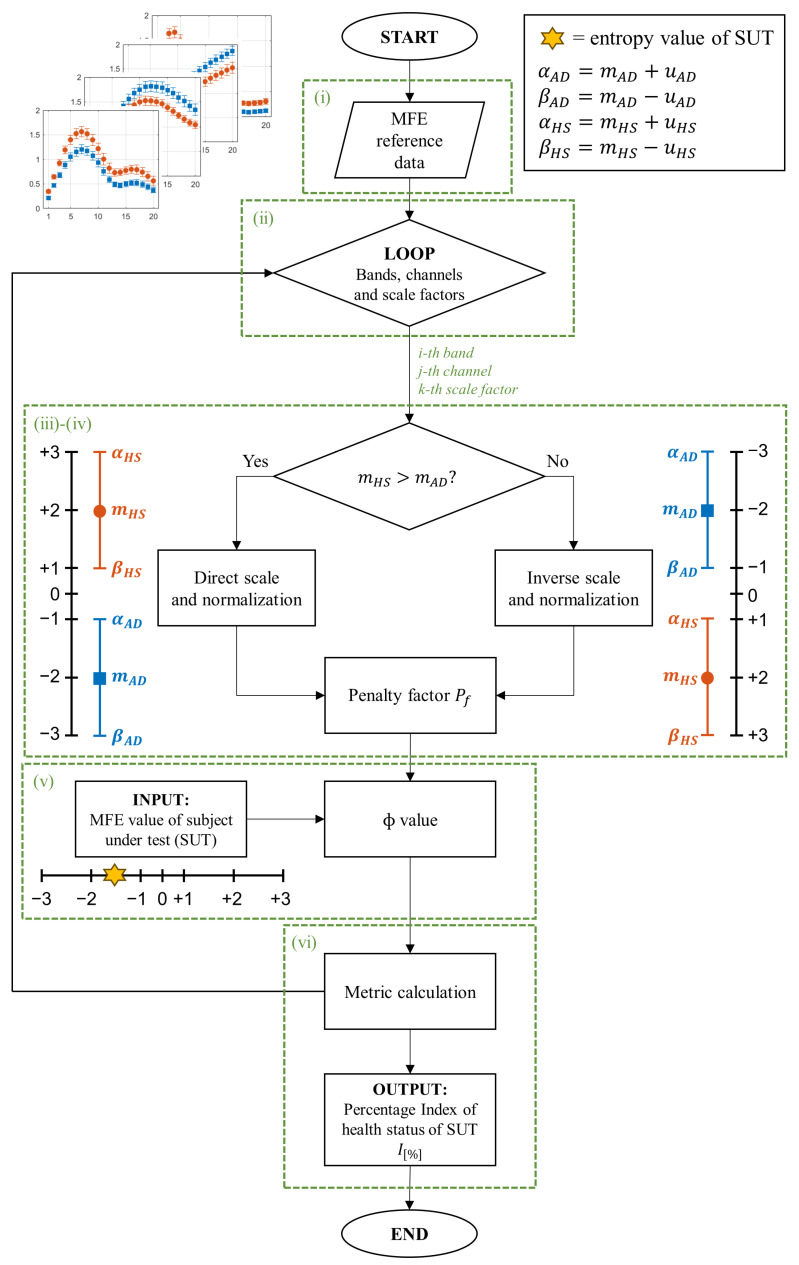
Flowchart of the algorithm developed to diagnose AD. $m_{AD}$: mean of MFE values for AD subjects, $u_{AD}$: uncertainty of MFE values for AD subjects, $m_{HS}$: mean of MFE values for HS subjects, and $u_{HS}$: uncertainty of MFE values for HS subjects.

The 80 EEG reference data (40 HS and 40 AD) were analyzed epoch-wise to calculate the MFE values, based on [9]. Specifically, the MFE parameters m and r were set to 2 and 0.20, respectively, and a range of scale factors from 1 to 20 was chosen. For each subject and each EEG channel, the MFE values in the specific frequency range (all, δ, θ, α, and β) were obtained by averaging across the 14 epochs. Figure 3 shows an example of the MFE values depending on the scale factors of the AD subjects and those of the HS of the reference dataset for channel P4 at the different EEG bands. As expected, the MFE curves exhibit a different behavior between HS and patients with AD, according to results obtained in our previous work [9] summarized in Table 1. Based on these considerations, two ranges of reference were developed, one related to the HS state and one related to the AD state.For each band, channel, and scale factor, the reference range was established by utilizing the mean values of MFE ($m_{HS}$ and $m_{AD}$) and the mean uncertainty ($u_{HS}$ and $u_{AD}$), calculated as follows:
(4)$$m=\frac{1}{N}\sum_{i=1}^{N}MFE_{i}$$
(5)$$u=\frac{1.96\;\cdot \;s}{\sqrt{N}}$$
where *N* is the number of subjects in the group, $MFE_{i}$ is the MFE value for the *i*th patient, 1.96 is the coverage factor to establish a 95% confidence interval, and *s* is the standard deviation of the reference data.A linear transformation of the original MFE data was adopted to report the MFE values on a standardized scale. In more detail, the empirical selection of the standardized scale considered the typical amplitude range for MFE values and the separability between HS and AD. As can be observed from Figure 3, the trends of MFE exhibit a maximum excursion of amplitude 2 for all the frequency bands. Consequently, it was deemed reasonable to maintain intervals of amplitude 2 in the standardized scale. A positive-value interval was assigned to depict the MFE range for healthy subjects, with +1 representing the lower bound and +3 representing the upper bound on the standardized scale. Conversely, a negative-value interval was designated for the MFE range in subjects with AD. Moreover, the intermediate zone was consistently selected with a width of 2. Hence, the extremes of the standardized intervals were chosen as follows:[+3,+1] represents the HS condition (with +2 corresponding to the $m_{HS}$);[−1,−3] represents the AD condition (with −2 corresponding to the $m_{AD}$);[−1,+1] is the intermediate zone.On the other hand, it is important to emphasize that the reference entropy trends in the HS and AD cases show opposite trends in some of the EEG bands, as reported in Table 1, and established in [14,29]. This phenomenon requires a distinction between the two cases, as illustrated in Figure 2, and a reorganization of the extreme values of the scale:If $m_{HS}>m_{AD}$, the scale range follows an ascending order, consistent with MFE values (direct scale);If $m_{HS}<m_{AD}$, the inverse scale range is in descending order, in contrast with MFE values (inverse scale).In addition to the distinction based on mean values, it is fundamental to take into account the impact of uncertainty. More specifically, intervals characterized by smaller uncertainty are more reliable than intervals with larger uncertainty because the former means that the reference data values are more consistent. In the worst cases, an overlap between the two reference ranges HS and AD may occur. Consequently, reliability and discriminative ability in that condition decrease. These considerations suggest the introduction of a factor that penalizes more significantly the cases in which entropy intervals are poorly separated or overlap. To this end, a so-called penalty factor, $P_{f}$, was defined according to the following equation:
(6)$$P_{f}=1+ln\left(u+ol+1\right)$$
where *u* represents the mean uncertainty, ol indicates the area of overlap between the intervals, and ln is the natural logarithm. By definition, $P_{f}$ takes on values greater than or equal to 1. In an ideal scenario with zero uncertainty (u=0) and no overlap (ol=0), the logarithmic term in Equation (Equation 6) is nullified, rendering $P_{f}$ equal to 1 and having no impact on the final output. In other cases, $P_{f}$ logarithmically increases in relation to finite uncertainty (u>0) and the overlapping area (ol>0).After defining the two reference ranges for AD and HS, and the $P_{f}$, SUTs were considered. Consequently, the MFE values for each band, channel, and scale factor were obtained and used to evaluate the status of the subjects. Then, an analysis is performed to determine the proximity of the test value to one condition over another, considering its placement within the scale.In this regard, a specific parameter, called the $\phi_{0}$ value, was introduced to quantify the degree of closeness to a specific condition. In more detail, it is computed for each scaling factor, channel, and band, according to the following formula:
(7)$$\phi_{0}=\frac{x_{SUT}-x_{min}}{x_{max}-x_{min}}\;\cdot \;\left(E_{max}-E_{min}\right)+E_{min}$$
where $x_{SUT}$ denotes a specific MFE value of the SUT, $x_{min}$ and $x_{max}$ are the extremes of the considered interval in which the $x_{SUT}$ falls (i.e., $\alpha_{HS},\beta_{HS},\alpha_{AD}\;or\;\alpha_{HS}$) and $E_{max}$ and $E_{min}$ represent the extreme of the corresponding direct or inverse standardized scale (ranging from −3 to +3). Therefore, $\phi_{0}$ scores correspond to the MFE values of the SUT translated into the standardized scale defined in phase (iii). Table 2 summarizes all the possible cases with the associated $\phi_{0}$.This value $\phi_{0}$ has to be appropriately weighed based on the previously mentioned considerations regarding the uncertainty and overlap of the MFE reference ranges.Recalling Equations (Equation 6) and (Equation 7), the final value is then calculated as follows:
(8)$$\phi =\frac{\phi_{0}}{P_{f}}$$As anticipated in phase (iv), in the ideal case where *u* and the ol values are equal to 0, the $P_{f}$ value is equal to 1, leaving the calculated value of ϕ unchanged. In the opposite case where either value is non-zero, $P_{f}$ increases logarithmically, reducing the value of the ϕ output and thus associating it with less weight in the calculation of the final metric.The final output of the procedure is obtained as the sum along scale factors, channels, and bands of all these weighed entropy values, which represent a comprehensive measure of the signal complexity and so can be used to discriminate between normal and pathological EEG signals. In more detail, a specific percentage index ($I_{\left[\%\right]}$) is calculated, which can be expressed as the probability that a subject has AD ($I_{AD[\%]}$), considering all the negative ϕ values to all entropy values, or as the probability that a subject is healthy ($I_{HS[\%]}$), considering all the positive ϕ values to all entropy values. The formulas are given below.
(9)$$I_{AD[\%]}=100\;\cdot \;\frac{\sum_{i=1}^{bn}\sum_{j=1}^{ch}\sum_{k=1}^{s_{f}}\phi_{i,j,k}^{-}}{\sum_{i=1}^{bn}\sum_{j=1}^{ch}\sum_{k=1}^{s_{f}}\phi_{i,j,k}}$$
where bn is the number of frequency bands (i.e., 5), ch is the number of channels (i.e., 19), $s_{f}$ is the number of scale factors (i.e., 20), $\phi_{i,j,k}^{-}$ as *i*, *j*, and *k* change represent all the negative values among the calculated ϕ values.
(10)$$I_{HS[\%]}=100\;\cdot \;\frac{\sum_{i=1}^{bn}\sum_{j=1}^{ch}\sum_{k=1}^{s_{f}}\phi_{i,j,k}^{+}}{\sum_{i=1}^{bn}\sum_{j=1}^{ch}\sum_{k=1}^{s_{f}}\phi_{i,j,k}}$$$\phi_{i,j,k}^{+}$ as *i*, *j*, and *k* change represent all the positive values among the calculated ϕ values. These two indices are not complementary in the sense that their sum does not always equal 100%. This discrepancy arises because the entropy values fall within the middle range. Consequently, considering both indices may prove useful, as will be discussed in Section 5.

## 5. Results and Discussion

As previously stated, upon establishing the reference intervals, the proposed metric was validated by exploiting different SUTs. Specifically, the validation dataset was composed of 12 HS and 12 AD subjects, selected from the two datasets as detailed in Section 4. For each SUT, the first step involved computing the MFE values. Subsequently, employing the Equation (Equation 8), the ϕ values for each scale factor, channel, and frequency band were obtained. Then, by applying the described procedure, the final percentage indexes of SUT’s health status can be obtained, according to Equations (Equation 9) and (Equation 10).

The outcomes for different SUTs are summarized in Figure 4, which reports the $I_{AD[\%]}$ (blue) and $I_{HS[\%]}$ (red) indices for each SUT. Specifically, Figure 4a depicts the indices obtained for HS, while Figure 4b displays the indices for AD. In the case of HS, the $I_{HS[\%]}$ index exceeds the threshold of 50% in the majority of instances. On the contrary, for patients with AD, the $I_{AD[\%]}$ index prevails. This underscores the capability of the proposed metric, derived from the complexity of the EEG signal, to effectively capture the state of the SUT.

To assess the efficacy of the proposed AD detection test, the accuracy, diagnostic odds ratio (DOR), and Matthews correlation coefficient (MCC) [30,31] were examined.

The classification accuracy, denoting the overall correctness of predictions (categorized as AD or healthy subjects), is defined as follows [30]:(11)$$Class.\;Accuracy=\frac{total\;correct\;predictions}{total\;number\;of\;predictions}$$
where totalcorrectpredictions refers to cases where AD exhibits a greater $I_{AD[\%]}$ and HS presents greater $I_{HS[\%]}$, whereas the totalnumberofpredictions is equal to the number of considered SUTs.

The DOR is defined as the ratio of the odds of a positive test result in subjects with the disease to the odds of a positive test result in subjects without the disease, as follows [30]:(12)$$DOR=\frac{TP\;\cdot \;TN}{FP\;\cdot \;FN}$$

Whereas the MCC is defined as follows [31]:(13)$$MCC=\frac{TP\;\cdot \;TN-FP\;\cdot \;FN}{\sqrt{\left(TP+FP)(TP+FN)(TN+FP)(TN+FN\right)}}$$

In Equations (Equation 12) and (Equation 13), TP=truepositives indicate AD with higher $I_{AD[\%]}$, TN=truenegatives represents HS with higher $I_{HS[\%]}$, FN=falsenegatives and FP=falsepositives are, respectively, AD patients for whom $I_{HS[\%]}$ prevails and HS for whom $I_{AD[\%]}$ prevails. The DOR ranges from zero to infinity; a higher DOR is indicative of better test performance [32] and it is familiar and easily interpretable for medical practitioners [33,34]. The MCC ranges between −1 and +1, with −1 for perfect misclassification (TP=TN=0) and +1 for perfect classification (FP=FN=0); MCC=0 indicates random classification (TPTN=FPFN).

Table 3 reports classification accuracy, DOR, and MCC obtained.

However, in addition to the final output value, it may be useful for the clinician to examine the trend of the ϕ value across different channels and bands as a preliminary assessment. The ϕ value helps to assess which bands and channels indicate the subject’s proximity to either AD or a healthy condition. In this regard, the ϕ values for different channels and frequency bands for HS#10 and AD#6, are reported in Figure 5 and Figure 6, respectively. As noted, for the AD patient, the ϕ values predominantly fall within the AD range across all bands, while for HS, they predominantly align with the HS range. This is reflected in the obtained indices.

It is possible to observe that the metric fails in some cases, specifically for HS#3 and HS#11, as well as AD#11 and AD#12 patients. Moreover, for HS#3 and HS#11, $I_{AD[\%]}$ prevails, while for AD#11 and AD#12, $I_{HS[\%]}$ is higher. Several factors may contribute to the failure of the metric. On the one hand, these shortcomings could be associated with the degree of AD, given the lack of information on the severity level of AD. Consequently, misidentifications may refer to subjects with low levels of AD. In contrast, the lack of information regarding the presence of the ApoEϵ4 allele, which is associated with AD, is notable for healthy subjects. According to the existing literature [35], the existence of these alleles in HS could be linked to the complexity of the EEG signal in various regions of the brain. These changes could be associated with anatomical alterations linked to neurodegeneration, potentially increasing the risk of developing dementia due to AD before its clinical onset.

### 5.1. Preliminary Findings from an MCI Case Study

As an exploratory extension of this study, the developed MFE-based algorithm was tested on an MCI subject. This test can be useful as an initial investigation to evaluate the method’s diagnostic capabilities for such a condition. The results show an index equal to $I_{AD[\%]}=$ 55.1%, reflecting the indeterminate nature of MCI as an intermediary stage between HS and AD. Furthermore, as can be seen in Figure 7, opposite trends in terms of ϕ values occur in different EEG frequency bands. Specifically, ϕ values are very high in the delta and alpha bands and remarkably low in the theta and beta bands, suggesting atypical brain complexity patterns that need further investigation. These findings necessitate a careful interpretation and point to the algorithm’s potential as a nuanced diagnostic tool, sensitive to various levels and causes of cognitive impairment. To validate these preliminary results, further research will involve a diverse cohort of MCI subjects, with different degrees and etiologies of cognitive decline.

### 5.2. Discussion

Based on the obtained findings, the proposed metric shows the potential to provide a cognitive assessment tool for detecting AD, offering several advantages. First, it presents an objective alternative to the subjective nature of the MMSE, providing a reliable means of evaluation. Moreover, the utilization of EEG signals ensures a cost-effective and non-invasive approach, setting it apart from traditional methodologies such as PET and MRI. Another aspect to underline is the efficiency of the metric proposed in the detection of AD using few data. In fact, it requires a few epochs of the EEG signal to evaluate the complexity and calculate the associated indices, $I_{HS[\%]}$ and $I_{AD[\%]}$. This is particularly advantageous compared to machine and deep learning methods, which require a large amount of data, often not available in this clinical field. Moreover, compared to advanced artificial intelligence approaches, the proposed method is more interpretable, facilitating comprehension by medical personnel.

However, it is worth highlighting some current limitations of the proposed metric. First, for optimal generalization of the metric, it could be necessary to incorporate a larger and more diverse cohort in the establishment of reference ranges. This should encompass considerations of age and gender among subjects to enable the derivation of normative references for medical professionals. Notably, the absence of normative references regarding complexity emphasizes the significance of this effort. Furthermore, improving the metric’s applicability for early detection necessitates the incorporation of varying degrees of AD, including MCI. An additional avenue for improvement lies in exploring the analysis of complexity with eyes open and under stimulation. Indeed, olfactory stimulation may offer insights into the state of AD, given that the olfactory system is reported to be one of the earliest targets of β-amyloid plaques and Tau tangles [36]. As a final aspect, it should be noted that the present study has concentrated on diagnosing AD in isolation, without considering comorbidities. Although this may seem to be a negative aspect in some ways, this focused approach was adopted to validate the application of MFE as a diagnostic tool in cases where AD is the primary concern. The study’s methodology was designed to establish a foundational understanding of MFE’s utility in identifying AD. This provides a benchmark for future research that may include a broader spectrum of cognitive conditions.

## 6. Conclusions

The present study introduced a novel metric based on MFE for AD detection. Specifically, considering AD and HS patients as distinct distributions, the proposed approach involves defining two reference intervals based on entropy trends, incorporating both mean values and associated uncertainties with a 95% confidence interval from the respective distributions. These reference intervals serve as a basis for comparing the MFE values of a SUT. The proposed approach allows for the calculation of two percentage indexes, namely $I_{AD[\%]}$ and $I_{HS[\%]}$, indicating the proximity of the SUT to either the AD or HS condition. The findings obtained from applying this approach to public EEG datasets show the potential effectiveness of the proposed method as a new robust and objective tool for AD detection, leveraging MFE-based brain complexity analysis. However, it should be mentioned that in real-world clinical settings, patients often present with multiple overlapping symptoms, and distinguishing AD from other forms of cognitive decline is a nuanced challenge. In this regard, future works will aim to address these complexities by incorporating a wider range of diagnostic scenarios, including subjects with comorbidities and those with different AD severity levels. Efforts will be dedicated to improving the handling of inter-subject variability, which remains a key challenge given the heterogeneous manifestation of AD across individuals. Furthermore, collaborations with clinical experts and neurologists are planned to conduct a comprehensive experimental campaign, designed to validate and refine the proposed method. 

## Figures and Tables

**Figure 1 bioengineering-11-00324-f001:**
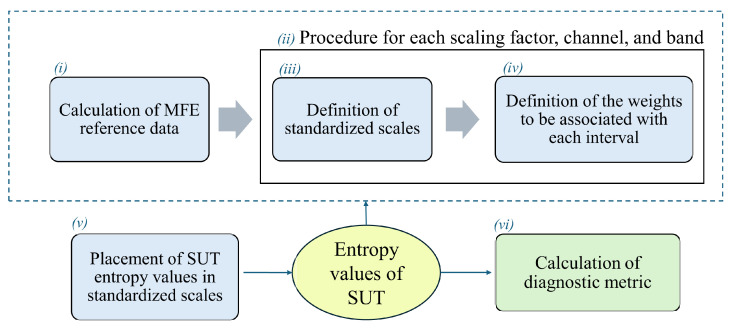
Pipeline of the proposed method for AD diagnosis.

**Figure 3 bioengineering-11-00324-f003:**
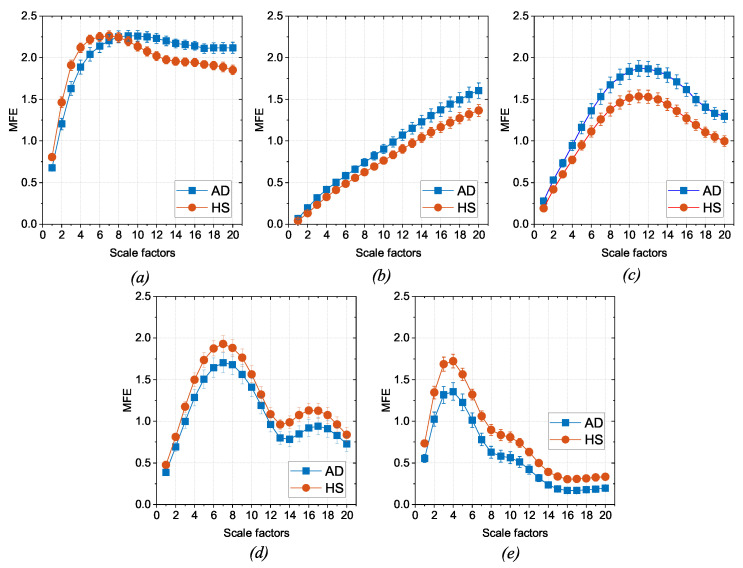
Comparison between the MFE reference values of the AD patients and HS, depending on the scale factors for the P4 channel. The results are reported in terms of the mean (dots) and uncertainty (bars): (**a**) all (0.5–30 Hz); (**b**) δ (0.5–4 Hz); (**c**) θ (4–8 Hz); (**d**) α (8–13 Hz); (**e**) β (13–30 Hz).

**Figure 4 bioengineering-11-00324-f004:**
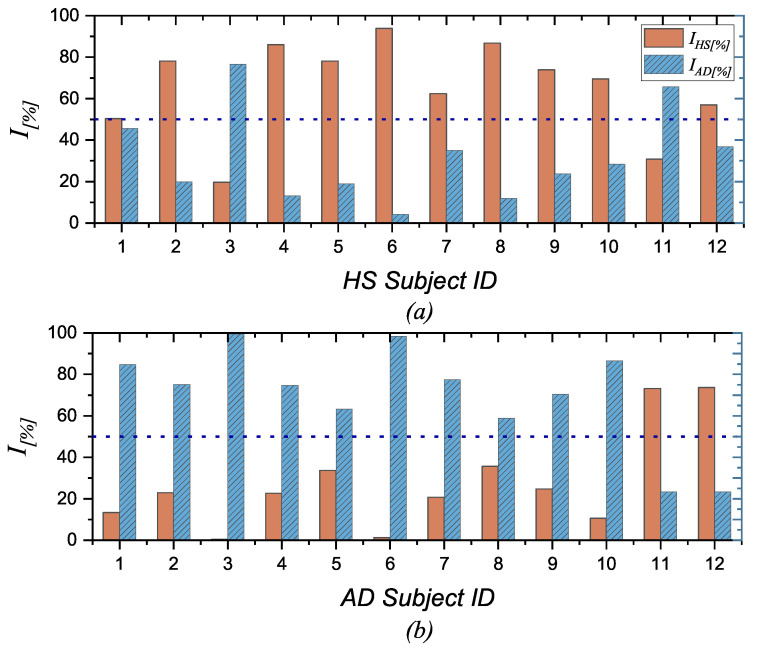
$I_{\left[\%\right]}$ for each SUT. (**a**) displays the HS subject, indicating $I_{HS[\%]}$ (in red) and $I_{AD[\%]}$ (in blue) for each of them. Notably, in this case, $I_{HS[\%]}$ is mostly higher than the threshold 50%; (**b**) reports the AD patient, presenting $I_{HS[\%]}$ (in red) and $I_{AD[\%]}$ (in blue) for each. It is observed that, in this case, $I_{AD[\%]}$ generally exceeds the 50% threshold.

**Figure 5 bioengineering-11-00324-f005:**
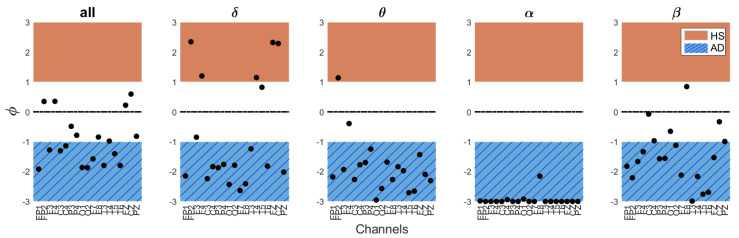
ϕ values across EEG bands and channels for subject #10 correctly diagnosed with AD, with an overall AD index $I_{AD[\%]}=$ 86.5%.

**Figure 6 bioengineering-11-00324-f006:**
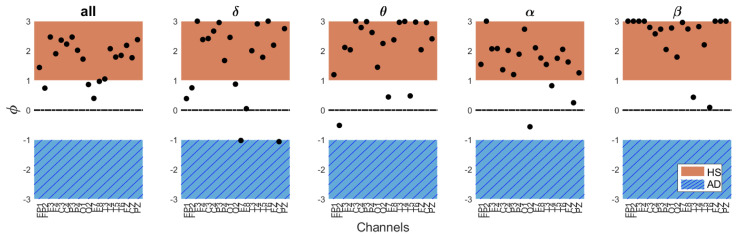
ϕ values across EEG bands and channels for subject #6 correctly identified as HS, with an overall HS index $I_{HS[\%]}=$ 93.9%.

**Figure 7 bioengineering-11-00324-f007:**
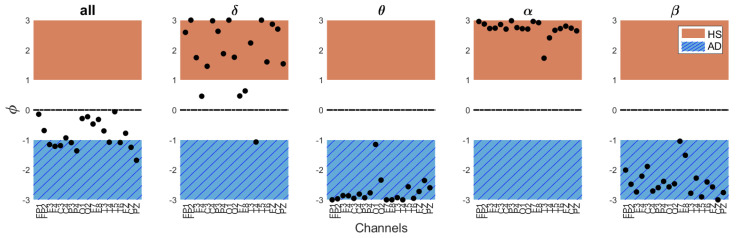
ϕ values across EEG bands and channels for the MCI subject, with an overall AD index $I_{AD[\%]}=$ 55.1%.

**Table 1 bioengineering-11-00324-t001:** Summary of the findings in [9].

Band	MFE	Notes
All	AD<HS on the short time scales AD>HS on the long time scales	At short time scales, AD brain activity shifts toward regularity losing physiologic complexity, while it exhibits a more chaotic and nonfunctional behavior at long time scales.
δ θ	AD>HS AD>HS	Cognitive decline is represented by a shift of brain activities to lower frequencies, implying higher complexity for AD subjects.
α β	AD<HS AD<HS	In this case, the shift in AD brain activity from higher to lower frequencies implies a loss of complexity compared to the HS group.

**Table 2 bioengineering-11-00324-t002:** Formulas of $\phi_{0}$ for each condition and interval.

	HS Interval	Intermediate	AD interval
$m_{HS}>m_{AD}$	$\frac{x-\beta_{HS}}{\alpha_{HS}-\beta_{HS}}\;\cdot \;\left(3-1\right)+1$	$\frac{x-\alpha_{AD}}{\beta_{HS}-\alpha_{AD}}\;\cdot \;\left(1+1\right)-1$	$\frac{x-\beta_{AD}}{\alpha_{AD}-\beta_{AD}}\;\cdot \;\left(-1+3\right)-3$
$m_{HS}<m_{AD}$	$\frac{x-\beta_{HS}}{\alpha_{HS}-\beta_{HS}}\;\cdot \;\left(1-3\right)+3$	$\frac{x-\alpha_{HS}}{\beta_{AD}-\alpha_{HS}}\;\cdot \;\left(-1-1\right)+1$	$\frac{x-\beta_{AD}}{\alpha_{AD}-\beta_{AD}}\;\cdot \;\left(-3+1\right)-1$

**Table 3 bioengineering-11-00324-t003:** Class. Accuracy, DOR, and MCC on 24 test subjects (12 HS and 12 AD).

Class. Accuracy	DOR	MCC
83%	25	0.67

## Data Availability

No data were acquired directly for this study. Requests for access to the two public datasets should be directed to the respective responsible parties.

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
