# Peer review of "A Novel Metric for Alzheimer’s Disease Detection Based on Brain Complexity Analysis via Multiscale Fuzzy Entropy"

_bioengineering, 2024, doi:10.3390/bioengineering11040324_

Round 1
Reviewer 1 Report
Comments and Suggestions for Authors
This paper presents a metric for Alzheimer’s disease (AD) detection based on brain complexity analysis by using Multiscale Fuzzy Entropy (MFE).
Section 3 should be expanded to give more details of the proposed method. What are m_{HS} and m_{AD} shown in Figure 1 and how are they calculated? Equations for their calculation should be given. The calculation of other variables shown in Figure 1 should also be given.
What is the definition of Multiscale Fuzzy Entropy (MFE) and how is it calculated? Some details should be given.
Author Response
The Authors would like to thank the Reviewer for reviewing the manuscript. In the following and the revised manuscript, the Authors have addressed all the considerations made by this Reviewer. We are confident that the overall quality of the work has been improved as a result of this review process. Please look at the attached pdf file for point-by-point answers

Reviewer 2 Report
Comments and Suggestions for Authors
The authors present a preliminary investigation of the feasibility of employing a measure of signal complexity, multiscale fuzzy entropy (MFE) applied to EEG data to discriminate between individuals with Alzheimer’s disease (AD) and healthy subjects (HS).
Previous investigations by the same authors and also by other investigators have demonstrated that MFE is abnormal AD. MFE is greater in AD, compared with HS, in the delta and theta frequency bands, but less in the alpha and beta band.
A recently published study by the same authors (reference 8) provides substantial evidence that the MFE has potential to discriminate between AD and HC. The richness of the EEG data (which varies across electrode site, frequency band, and time scale) indicates potential for discrimination. On the other hand, EEG signals are intrinsically very variable across time, and across many aspects of the data acquisition, raising concern about the reliability of measurements based on EEG.
In this manuscript that authors extend their previous work by employing a procedure to derive a composite measure derived from MFE estimates across electrode site, frequency band and time scale. They determine the distribution of values of this composite measure of MFE in HS and in AD in a training set of 40 HS and 40 AD cases. They propose a procedure for estimating the probability that the observed composite MFE measurement in a test subject lies within the expected range for HS and the also the probability that the observed MFE for the test subject lies in the range expected for a case of AD. They apply this to estimate the probability that each of 24 test subjects (12 HS and 12 AD cases) is either a HS or an AD case. They achieved 83% classification accuracy. One of the 12 HC was mis-classified as AD and 2 of the AD cases were mis-classified as HS.
The EEG data was derived from two separate data banks (one from Temple University, Philadelphia, USA; the other from Chung-Ang University Hospital, Seoul, Republic of Korea). The authors provide no information about the severity of the dementia in the AD cases. Therefore it is difficult to judge whether or not the reported classification accuracy is potentially a clinically useful degree of classification. The distinction between severe dementia and healthy status using routine clinical observations (history and mental state examination is a trivial challenge, and this classification performance would not be impressive in such a sample
In clinical practice, the major diagnostic changes in distinguishing between early stage AD and other causes of cognitive and functional impairment, such as depression.
On the basis of previously published work, especially the authors own work [ref 8] based on a sample of cases that appears to be partially overlapping with the sample in this current manuscript, I consider that MFE has potential to contribute to the diagnosis of AD. However, it is difficult to judge whether or not this manuscript adds appreciably to the preexisting level of confidence in the utility of MFE for diagnosing AD.
I consider that it is essential that the authors provide a better characterisation of the sample of cases , preferably with some estimate of the severity of dementia in both the training data set and in the test data set.
In light of the likelihood that diverse sources of noise contribute to the estimate of MFE, it would be helpful if the authors provided information about similarities and differences in the estimates of MFE from the two different data banks.
Furthermore in the discussion of the potential clinical utility of MFE they should acknowledge that in clinical practice, the challenging question is distinguishing AD for other cases of cognitive an functional impairment such as depression.
Author Response
The Authors would like to thank the Reviewer for his/her thorough review. In the revised version of the manuscript, the Authors have addressed all the issues raised by the Reviewer. The Authors believe that the overall quality of the work has benefited from this constructive review process. Please look at the attached pdf file for point-by-point answers

Round 2
Reviewer 1 Report
Comments and Suggestions for Authors
The authors have adequately addressed my comments.
Reviewer 2 Report
Comments and Suggestions for Authors
The authors have adressed my previous concerns in an adequate manner